# Metabolic Reprogramming in Tumor-Associated Macrophages in the Ovarian Tumor Microenvironment

**DOI:** 10.3390/cancers14215224

**Published:** 2022-10-25

**Authors:** Sudhir Kumar, Sonam Mittal, Prachi Gupta, Mona Singh, Pradeep Chaluvally-Raghavan, Sunila Pradeep

**Affiliations:** 1Department of Obstetrics and Gynecology, Medical College of Wisconsin, Milwaukee, WI 53226, USA; 2Department of Physiology, Medical College of Wisconsin, Milwaukee, WI 53226, USA; 3Medical College of Wisconsin Cancer Center, Medical College of Wisconsin, Milwaukee, WI 53226, USA

**Keywords:** tumor-associated macrophages (TAMs), tumor microenvironment (TME), metabolism, metabolic reprogramming, ovarian cancer, extracellular vesicles (EVs)

## Abstract

**Simple Summary:**

The highly metastatic and immunosuppressive microenvironment of ovarian cancers is a major determinant of the aggressive nature and therapeutic resistance of ovarian cancer. Therefore, we believe that a thorough understanding of the mechanisms that regulate the composition and function of the tumor microenvironment is critical for the development of a more effective course of treatment for this devastating malignancy. This review summarizes the recent literature on the major metabolic pathways affecting macrophage immune metabolism and its impact on phenotypic and functional changes in macrophages in the ovarian tumor microenvironment.

**Abstract:**

The interaction between tumor cells and macrophages in the tumor microenvironment plays an essential role in metabolic changes in macrophages and reprograms them towards a pro-tumorigenic phenotype. Increasing evidence indicates that macrophage metabolism is a highly complex process and may not be as simple as previously thought. Pro-inflammatory stimuli switch macrophages towards an M1-like phenotype and rely mainly on aerobic glycolysis and fatty acid synthesis, whereas anti-inflammatory stimuli switch macrophages towards an M2-like phenotype. M2-like macrophages depend more on oxidative phosphorylation (OXPHOS) and fatty acid oxidation. However, this metabolically reprogrammed phenotypic switch in macrophages remained a mystery for a while. Therefore, through this review, we tend to describe how macrophage immunometabolism determines macrophage phenotypes and functions in tumor microenvironments (TMEs). Furthermore, we have discussed how metabolic reprogramming in TAM can be used for therapeutic intervention and drug resistance in ovarian cancer.

## 1. Introduction

Ovarian tumor development is a complex, heterogeneous process influenced by several epigenetic and microenvironmental factors [1,2]. A poor diagnosis and high relapse rate make ovarian cancer one of the deadliest gynecological malignancies compared to other cancers in females [3]. Studies have shown that tumor microenvironments (TMEs) that include the tumor mass, extracellular matrixes, secretory factors, resident cells, invading cells, and immune cells play an essential role in tumorigenesis and can also serve as therapeutic targets [4,5,6]. At advanced stages, ovarian cancer cells invade the omentum from the peritoneal cavity, and during dissemination, cancer cells leave the primary site and spread to the highly metastatic TME. Stromal and immune cells assist in this process. In ovarian cancer, among various immune cells, macrophages are one of the most abundant cell populations [1]. The growing information indicates that macrophages potentially influence ovarian cancer growth and metastasis [7]. Furthermore, ovarian tumor cells and macrophages utilize a sophisticated signaling network that connects malignant and non-malignant cells to take advantage of the hypoxic tumor microenvironment [8]. The macrophages in the TME constitute a fraction of the cancer’s inflammatory microenvironment and are termed tumor-associated macrophages (TAM) [9]. Adaptive anti-tumor immune responses and TME-associated cytokines are responsible for shaping TAMs function [10]. Its abundance and high degree of plasticity in the ovarian tumor microenvironment provide a battlefield for the immune cells to fight against cancer [11]. TAMs are involved in several ways, including ovarian cancer patient prognosis, chemoresistance, inflammation, metabolism, and metastasis [12]. Traditionally, TAMs are classified as M1 and M2 macrophages, but this binary classification of macrophages is difficult due to their diverse nature. The recent technological advances have led to pro- and anti-tumorigenic TAM being classified as M2a, M2b, and M2c instead of M2 or more permissive terminology such as M1- or M2-like macrophages [13]. Therefore, a better understanding of the crosstalk between tumor cells and macrophages in the TME is necessary for the development of novel therapeutics and to overcome chemoresistance against ovarian cancer [14].

Recent evidence suggests that macrophages are highly reprogrammed in TMEs [15,16]. Like other solid tumors, ovarian cancer in TMEs preferentially opts for glycolysis over OXPHOS and carries out metabolic transcriptional reprogramming to adapt in adverse conditions, which directly represents the ‘Warburg effect’ [17]. In the TME, cancer cells produce lots of metabolic waste such as lactic acid [18]. The interaction between ovarian cancer cells and TAMs leads to metabolic competition in tumor TMEs, restricts the nutrients required by TAMs, and leads to microenvironmental acidosis, which causes TAMs to undergo metabolic reprogramming and affects the progress of ovarian cancer [19,20]. The metabolic processes such as glycolysis, tricarboxylic acid (TCA), pentose phosphate pathway (PPP), arginine, glutamine, and fatty acids metabolism precisely affect the function of macrophages [21,22]. Hence, manipulating these metabolic pathways can dramatically change the macrophage functioning in specific ways rather than just generating energy or general biosynthesis, which will help combat ovarian cancer. Metabolic reprogramming is thus combined as a phenomenon with other important immunoregulatory events that govern health and illness [23,24]. This article aims to review the current knowledge of metabolic changes associated with TAM regarding disease progression and treatment of ovarian cancer.

## 2. Metabolic Changes in TAMs

Tumor-infiltrating macrophages or macrophages residing in ascites may alter their metabolism in the tumor microenvironment [19]. TAMs with unique glucose, lipid, amino acid, oxygen, and iron consumption support their pro-tumor and immunosuppressive behavior [25]. In experimental settings, it has been shown that metabolic pathways may be targeted to promote macrophage conversion in either a pro- or anti-tumoral way in a specific tumor microenvironment [26]. Historically, M1 and M2 have been classified based on arginine metabolism. Besides the arginine metabolism, several other metabolic pathways reprogram the TAMs in the TME. Several soluble and insoluble components in the ovarian TME, including the peritoneal and primary sites, strongly influence the polarization of TAMs towards either an M1- or M2-like phenotype (Figure 1). The tumor microenvironment of ovarian cancer is very distinct from other solid tumors; the cancer cells can be easily shed from the primary tumor into the peritoneal cavity, forming malignant ascites [27]. Ascites is a primary tumor microenvironment site where different types of cells and metabolites crosstalk to play a significant role in reprogramming. Out of all diverse populations of cells, macrophages are a crucial player in the formation of malignant ascites [28,29]. The overlapping metabolic reprogramming of macrophages and cancer cells is one of the factors determining anti-tumor responses.

### 2.1. Glucose Metabolism

Glycolysis is linked with TAM recruitment and tumorigenesis [30]. TAMs, in response to chemo-attractants, such as cytokines, chemokines, and pro-inflammatory signals, infiltrate tumor sites [16]. A gradual reduction in oxygen through the mTOR pathway during extravasation and neo-angiogenesis are characteristics of TAM, which [31] depends on glycolysis because inhibiting macrophage glycolysis by dichloroacetic acid substantially decreases macrophage migration [32]. Studies have also shown that TAMs modulate oxidative phosphorylation (OXPHOS) and fatty acid oxidation (FAO) metabolism to attain the M2 phenotype in low glucose TMEs to provide an immune suppressive environment [33,34]. Therefore, TAM metabolism can force cancer cells to adopt glycolysis as their primary metabolic pathway, thus rendering an invasive cancer cell phenotype [16]. TAMs lead to metabolic cooperation with cancer cells and the establishment of a pro-invasive TME. However, further investigations into the contribution of TAM glycolysis in the recruitment and function of other immune and stromal cells are required. TAMs retain their glycolytic phenotype and migration, albeit of the TME’s hypoxic and nutritional restriction [35]. Additionally, critical glycolytic enzymes such as hexokinase 2 (HK2), phosphofructokinase, and enolase 1 (ENO1) were upregulated in the tumor model. TAM glycolysis is also linked to angiogenesis and tumor metastasis [35].

Hypoxic TAMs show a markedly higher glycolysis rate and an increase in growth factors, such as vascular endothelial growth factor (VEGF) and platelet-derived growth factor (PDGF), which aid the progressiveness of cancer by enhancing tumor growth, angiogenesis, and metastasis [36]. In other cancers such as lung cancer, alteration of the mTOR-REDD1 axis in hypoxic tumor cells reprograms the metabolism to a greater glycolytic state, resulting in reduced metastatic burden. Because TAMs themselves can induce tumor hypoxia and glycolysis in cancer cells, it may be possible that they help to activate 5′-adenosine monophosphate-activated protein kinase (AMPK), which then helps to increase the number of available glucose molecules and increase glycolysis flux [31,37]. As a result, the mitochondria in TAMs will experience an increase in oxygen consumption rate, resulting in more significant tumor hypoxia. As TAM metabolism causes cancer cells to turn to glycolysis as their main metabolic route, tumor cells acquiring the invasive phenotype are automatically forced to take up glycolysis [38]. TAMs through lysophosphatidic acid (LPA) are known to induce pseudohypoxia in ovarian cancer cells through HIF1α via Gαi2, Rac1, and NOX2 axis, which results in upregulation of the glycolytic enzyme hexokinase-2 (HK2) and glucose transporter-1 (GLUT1). This results in LPA-mediated glycolytic shift in EOC cells by TAMs [39].

Hence, TAMs are known for establishing metabolic coordination with cancer cells and creating a pro-invasive TME. Recently, eight glycolysis-related genes (ACTN3, ESRRB, DCN, etc.) were also identified, which are known as prognostic genes as they are the signature survival genes in ovarian cancer [40].

Despite these results, more research is needed to ascertain the role of TAM glycolysis in immune and stromal cell recruitment and function. The high glycolytic rate of tumor cells increases lactic acid production. Tumor cell-derived lactic acid induces hypoxia-inducible factor (HIF)-1α-dependent peritumoral polarization of TAMs, and hypoxia is associated with the accumulation of peritumoral TAMs [41]. Lactic acid induces VEGF production in TAMs by stabilizing HIF-1α and promoting neovascularization. Moreover, tumor-derived lactic acid activates mTORC1 to suppress ATP6V0d2-targeted HIF2α degradation in TAMs, leading to M2 polarization with enhanced HIF2α-mediated VEGF production [42]. Additionally, lactic acid produced by tumors stimulates IL-23 production in TAMs, resulting in tumor growth by inducing the production of IL-17 and IL-22 [36].

Altogether, the acidic TME results from high glycolysis of tumor cells, and poor perfusion plays a pivotal role in tumor progression. Acidity (independent of lactic acid) augments the pro-tumoral polarization of TAMs in prostate cancer [43]. In TAMs, tumor cell-derived lactic acid affects glucose metabolism. Besides the effect of tumor cell-derived lactic acid, enhanced endogenous aerobic glycolysis, also known for induction of pro-tumoral TAMs, was confirmed by a proteomic analysis illustrating the upregulation of HK2, phosphofructokinase, and ENO1 in TAMs.

### 2.2. Fatty Acid and Lipid Metabolism

Apart from glucose metabolism, lipids are a crucial metabolic hallmark of cancer. TAMs exhibit alterations in lipid metabolism, including increased fatty acid production, absorption, and storage, which have been linked to functional reprogramming, but the underlying processes remain largely not understood in ovarian cancer [44].

In the tumor microenvironment, adipocytes are the prime source of fatty acids. In the TME, adipocyte-derived lipids, including fatty acids, affect cancer cells and various peripheral cells, such as cancer-associated fibroblasts, dendritic cells, macrophages, and other immune cells [45]. Cancer cells and adipocyte interaction stimulate inflammatory cytokines closely related to lipid production. Yu et al. demonstrated that interleukin-17A (IL-17A), a pro-inflammatory cytokine, promoted the growth and metastasis of ovarian cancer by regulating fatty acid metabolism in adipocytes, primarily regulating fatty acid uptake by cancer cells [46].

Lipid metabolism plays a crucial role in TAMs differentiation and function. Recent reports suggested that macrophages show more lipid accumulation and dependency on fatty acid oxidation (FAO) compared to glycolysis in TMEs. Inhibition of FAO reduced the number of TAMs in the TME [47]. In addition, TAMs accumulated with polyunsaturated fatty acid (PUFA) and linoleic acid (18:2) demonstrate pro-tumoral effects on human ovarian carcinoma. The nuclear receptor peroxisome proliferator-activated receptor β/δ (PPARβ/δ) is a lipid ligand-inducible transcription factor associated with macrophage polarization [48]. High concentrations of polyunsaturated fatty acids, particularly linoleic acid, act as potent PPARβ/δ agonists in macrophages. These fatty acid ligands accumulate in lipid droplets in TAMs, providing a reservoir of PPARβ/δ ligands. These observations suggest that the deregulation of PPARβ/δ target genes by ligands of the tumor microenvironment contributes to the pro-tumorigenic polarization of TAMs associated with ovarian cancer. Lipid droplet formation serves as a pool to enrich TAMs with PPARβ/δ ligands, leading to the upregulation of PPARβ/δ target genes and the polarization of TAMs to a pro-tumoral phenotype in ovarian carcinoma [48].

Ovarian cancer cells are also responsible for the depletion of lipids rafts and the efflux of cholesterol from macrophages which promotes IL4-mediated reprogramming of macrophages by inhibiting the expression of IFNγ induced genes, hence promoting the tumorigenic functions of TAMs. Inhibiting the cholesterol efflux by deleting the ABC transporter reduces tumor progression and can be a novel therapeutic strategy [49].

Lipogenesis and lipid transport mediated decreased infiltration of M1 macrophages are the reason for the high metastatic rate in obese people [50]. It has been found that inhibition of lipid transporters such as fatty acid binding protein 4 (FABP4) significantly reduces chemoresistance and metastasis in ovarian cancer [51]. Studies have also shown the role of unsaturated fatty acids (oleate) in polarizing the bone marrow-derived myeloid cells to immunosuppressive TAMs [52].

Ovarian tumor cells are also found to release metabolites such as 5-lipoxygenase (5-LOX), a member of the lipoxygenase gene family, which is a crucial enzyme assisting in the conversion of arachidonic acid to 5-HETE and leukotrienes. Increased 5-LOX metabolites from hypoxic ovarian cancer cells promoted migration and invasion of macrophages, which is mediated by the upregulation of matrix metalloproteinase (MMP)-7 expression through the p38 pathway [53]. This increased MMP7 in TAMs is correlated with the enhanced generation of TNF-α and heparin-binding epidermal growth factor-like growth factor. Zileuton, a selective 5-LO inhibitor, potentially reduces the expression of MMP-7 and the number of infiltrating macrophages and can hence be used as a therapeutic target (47).

In ovarian cancer, TAMs regulate metabolic function through PPARβ/δ and some signature genes (e.g., LRP5, CD300A, MAP3K8, and ANGPTL4) associated with immune regulation and tumor progression that correlate with short relapse-free survival in serous ovarian cancer [44,48]

Several studies have shown that targeting lipid metabolism significantly reduces tumor burden. Studies have shown that targeting cholesterol metabolism using a simvastatin-based nanomedicine strategy reverses epithelial–mesenchymal transition and repolarizes TAM to treat drug-resistant cancer cells [54].

### 2.3. Amino Acid Metabolism

Several studies explored the potential consequences of amino acid metabolism on the functional reprogramming of TAMs. However, most of this research is observational, with only a few providing mechanistic insight. Arginine depletion in culture medium was the first proof of this theory in the 1970s, where it was demonstrated that macrophages could inhibit lymphocyte activity by depleting arginine in media [55]. The complex arginine metabolism is directly related to TAM’s reprogramming. The inducible nitric oxide synthase (iNOS) converts arginine into NO and L-citrulline. Arginine is a precursor for synthesizing polyamines, urea, glutamate, and other metabolites [35,56]. The NO generated inhibits OXPHOS by inhibiting enzymes involved in the TCA and electron transport chains and upregulates glycolysis. Its concentration is critical in regulating the macrophage’s inflammatory or anti-inflammatory response to maintain the homeostasis between the body’s defense and tissue damage.

Glutaminolysis, a process by which cells convert glutamine into the TCA cycle, has been correlated with ovarian cancer aggressiveness [57]. Recently, this process has been explored with the M2-like phenotype of macrophages in the tumor. In ovarian tumor, cells become addicted to extracellular glutamine when they are in a high glutaminolysis state [58]. Glutamine addiction elicits crosstalk between cancer cells and macrophages, where cancer cells release N-acetyl aspartate (NAA), resulting in enhancing glutamine synthetase (GS) expression-mediated M2 polarization of macrophages [58].

Apart from this, some amino acids such as aspartic acid, asparagine, branched-chain amino acids (Valine, Leucine, Isoleucine), and some 1 carbon (1C) metabolites (cystine, pyrimidines, thymidine, S-adenosylmethionine, and glutathione, etc.) also play a role in ovarian cancer metabolic reprogramming. However, specific mechanisms have yet to be explored in the context of TAM reprogramming.

### 2.4. Metabolic Reprogramming by Metabolite Shuttling between TAM and Cancer Cells

So far, we have discussed that metabolic reprogramming is a common feature of macrophages and tumor cells that function symbiotically. Moreover, macrophages directly ‘fuel’ tumors by exporting different metabolites preferentially used by adjacent tumor cells. Recently, Goossens et al. showed that cancer cells promote the efflux of cholesterol from macrophages, which are associated with increased IL4 expression, while decreased IFNγ-related gene expression results in reprogramming pro-tumorigenic TAMs [49].

It has also been reported that extracellular vesicles (EVs) released by tumor cells and macrophages participate in the metabolite shuttling in the tumor microenvironment [6]. However, little is known about their metabolite cargo and function in recipient cells, making this field a new area of vesicle research. EVs loaded with different metabolites can induce metabolic reprogramming in the recipient cells. For example, EVs isolated from cancer-associated fibroblasts carry vital metabolites, such as amino acids and TCA-cycle intermediates, which can be internalized by cancer cells under nutrient-deprived conditions in the tumor microenvironment [59]. The uptake of these metabolites by cancer cells can induces metabolic alteration that supports tumor growth and metastasis [59]. Many researchers are trying to utilize these metabolites as a potential biomarker in several diseases, as metabolites of EVs from human body fluids already represent a goldmine of tumor biomarkers [60,61,62].

Despite the various advancement in EVs-related research in the past few years, the question of the significance of EVs-mediated metabolic reprogramming in ovarian cancer is still unsolved. The following factors limit this study:The most effective metabolite in EVs that regulate metabolic reprogramming has not been identified yet.EVs’ content greatly resembles the cells of their origin; however, it is essential to note that EVs’ metabolite cargo is greatly affected by current cell culture conditions that sometimes generate EVs with distinctively different metabolites than their parent cells [63].It is unclear whether genes or EVs induce differences in metabolic profile in particular cells from other cells.There are still technical obstacles in isolating and purifying EVs from multiple body fluids for metabolite characterization [64].

In addition to these factors, the metabolite composition of EVs is highly affected by pre-analytical factors that include storage, handling, and quantification [64]. Enzyme activity and sample contamination also influence the metabolic profiling of EVs. All these factors contribute to the heterogeneity of experimental outcomes in EV research, thus decreasing the reproductivity of a method. It is important to note that studying one or a few molecules in a subgroup of EVs gives partial information on the functional role of the studied EV population. In contrast, analyzing whole metabolite profiling in a large subtype of EVs would provide more diverse and relevant results.

However, the lack of a consensual approach to the separation of EVs is a major obstacle to the advancement of EV research. Thus, further research should focus on developing standard methods for the isolation and purification of EVs and exploring the possibility of targeting the bioactive molecules in EVs to elucidate further the activities and interactions of EVs and the immunotherapeutic roles they may be able to play in the metabolic reprogramming of the TME.

### 2.5. TAM in Anti-Cancer Drug Resistance

The persisting challenge in clinical oncology is to improve cancer treatments’ efficacy. At the same time, the tendency of cancer cells to become resistant to the majority of treatments limits complete pathological regression [65]. A tumor’s therapeutic sensitivity depends on a complex interplay between cancer cells and the tumor microenvironment, especially the interaction between the cancer cells and immune cells. TAMs’ impact on tumor development varies according to tumor type, tumor microenvironment, and TAMs’ location according to intratumoral compartments. Reprogramming the macrophage phenotype into the pro-inflammatory to anti-tumor phenotype is an excellent strategy for cancer therapy. The tumor microenvironment influences both tumor-resident and infiltrating macrophage development, which may result in one of two different outcomes: tumor-specific macrophages or non-specific macrophages.

TAMs promote tumor development in solid tumors, increase angiogenesis, induce lymph angiogenesis, remodel the stroma, metastasize, and inhibit the immune system by modulating and releasing numerous pro-angiogenic factors, including VEGF-A, tumor necrosis factor (TNF), FGF, thymidine phosphorylase, urokinase plasminogen activator, and adrenomedullin (ADM). They also promote stromal remodeling, tumor cell invasion, and metastasis by releasing many enzymes, including plasmin, MMPs, cathepsin B, PDGF, and TGF-β1. TAMs have been shown to exhibit tumor-supporting properties in various cancers [66].

However, a growing number of experimental and clinical data suggest that TAMs play a dual function in tumor growth and survival, and their anti-cancer activity has been partially discovered [67]. However, TAMs pro- or anti-cancer activity depends on the tumor’s origin site [68]. Among the mechanisms by which TAMs promote tumor growth after chemotherapy are increased recruitment of immunosuppressive TAMs, pro-tumor polarization, activation of a tumor-promoting Th17 response, and activation of anti-apoptotic programs in malignant cells. Combining TAM depletion and chemotherapy resulted in a 50% reduction in tumor-vessel density. TAM reduction in tumor mass may normalize the vasculature by skewing perivascular TAMs away from pro-angiogenic cells and toward angiostatic cells, resulting in improved blood flow and drug delivery to tumors, thus encouraging tumor eradication. VEGF-A attracts macrophage progenitor cells, which differentiate into TAMs (M2 macrophages) in the presence of IL-4 [69]. Elimination of these macrophages has been shown to decrease tumor development, angiogenesis, and invasion. In chemotherapy-treated malignancies, lymphangiogenesis may be inhibited by blocking the VEGF-C/VEGFR3 axis. It must be apparent when it comes to cancer if cancer therapy activates TAMs or trains TAMs to foster tumor removal and metastasis. Recent investigations have shown that macrophages contribute to drug resistance and recurrence following chemotherapy treatment by promoting tumor revascularization, inhibiting cytotoxic T-cell immunity, and activating anti-apoptotic processes in cancer cells [70].

### 2.6. TAM as a Potential Metabolic Therapeutic Target

So far, it has been shown that tumor cell-derived stromal cells (which may be found in the tumor microenvironment) help the cancer cells proliferate, invade, and metastasize [10,71,72]. TAMs, as a critical regulator of the complex TME, have integrated actions to support tumor cell survival, increase tumor angiogenesis, and assist in tumor dissemination and spread. TAMs are a promising target for cancer therapy because of their ability to either induce or suppress tumor growth [73,74].

Current TAM therapies fall into three categories: (1) suppressing pro-tumor TAMs, which includes TAM recruitment inhibition and depletion; (2) activating anti-tumor TAMs, which refers to converting pro-tumorigenic macrophages to anti-tumorigenic macrophages; and (3) reprogramming of TAMs [74,75]. In a generalized notion, TAMs may be either tumor-killing (M1) or tumor-promoting (M2) [76]. Distinct macrophages and their accompanying characteristics show that different populations do exist. The transcriptomic studies of macrophages show that most of the total macrophage population exhibit M1 and M2 markers [10,70]. Regardless of the continuing debate over M1 and M2 polarity, the therapeutic issue remains to prevent macrophage trophic phenotypes (with their immunosuppressive properties and tumor growth-promoting properties) from occurring and supporting activation and anti-cancer activity [77]. Recent research has shown that a system such as this is practical and perhaps helpful [78,79].

One therapeutic approach based on macrophages is converting TAMs into M1 type macrophages. In this regard, OXPHOS pathways play a vital role in the M1 transition of M2 TAMs. Inhibition of mitochondrial OXPHOS is known for preventing the reprogramming of pro-inflammatory M1 macrophages to the anti-inflammatory M2 phenotype, reducing tumor burden. This was further supported by a study where metformin and IACS010759-mediated inhibition of OXPHOS has reduced the tumor burden [80]. Interestingly, studies have shown that combinatorial treatment with metformin and the glycolytic inhibitor 2-deoxyglucose (2-DG) is effective in a broad spectrum of pre-clinical cancer models [81]. More recently, metformin has been shown as a drug of choice in ovarian cancer as it targets multiple pathways, and it has been associated with better progression-free survival (PFS) and recurrence-free survival (RFS) in ovarian cancer [82]. Recently, metformin has also been found to target SPHK1, an enzyme that helps synthesize sphingosine-1-phosphate (S1P) [83,84]. S1P was recently identified as an immunosuppressive lipid in the ovarian cancer microenvironment [84]. Besides this, succinate; an oncometabolite produced by cancer cells promotes TAM polarization and metastasis [85]. Succinate dehydrogenase (SDH) is a mitochondrial metabolic enzyme complex involved in the electron transport chain and the citric acid cycle. SDH has been implicated as a tumor suppressor in ovarian cancer [86]. It has been observed that the knockdown of SDHB (succinate dehydrogenase subunit B) results in EMT in mouse ovarian cancer cells and histone hypermethylation, leading to metabolic vulnerability. In addition, by analyzing mitochondrial function, it was observed that SDH dysfunction leads to a decreased mitochondrial reserve capacity that can be exploited by energy stress caused by either glucose withdrawal or metformin treatment.

Therefore, it is worth exploring the possibility that tumors with SDH deficient mutations and SDH subtypes can be therapeutically targeted by exploiting their metabolic state. Recently, succinate dehydrogenase overexpression in ovarian cancer was targeted using the anti-metabolic compound shikonin, which demonstrated potent anti-tumor efficacy [87].

## 3. Conclusions

The recent advancements in ovarian cancer research suggest the unresponsiveness of the current therapeutic approaches in reducing mortality rates. It is becoming more evident that the metabolic reprogramming of the TAM and TME plays a crucial role in the tumor. According to recent studies, TAM enhances the progression of ovarian tumors, but because of its high degree of plasticity, it can also be used to suppress tumor growth by remodeling the M2-like phenotype into an M1-like phenotype, and the plasticity of TAMs can be harnessed in ovarian cancer therapeutics.

## 4. Future Perspective

Exploring the different states of TAMs using high throughput next-generation techniques such as single-cell sequencing and metabolomics research will improve the current understanding of ovarian TMEs and provide specific information about them. So far, the mechanistic insights gained from the molecular pathways required for these alterations have been limited, and thus current treatments that seek to modify metabolic pathways will be insufficient unless all therapeutic targets are explored. Considering TAMs as a key player will improve current regimen decision-making and open new avenues for precision immuno-oncology. In addition, macrophages can help with tumor therapy and the prevention and management of adverse effects in ovarian cancer treatment.

## Figures and Tables

**Figure 1 cancers-14-05224-f001:**
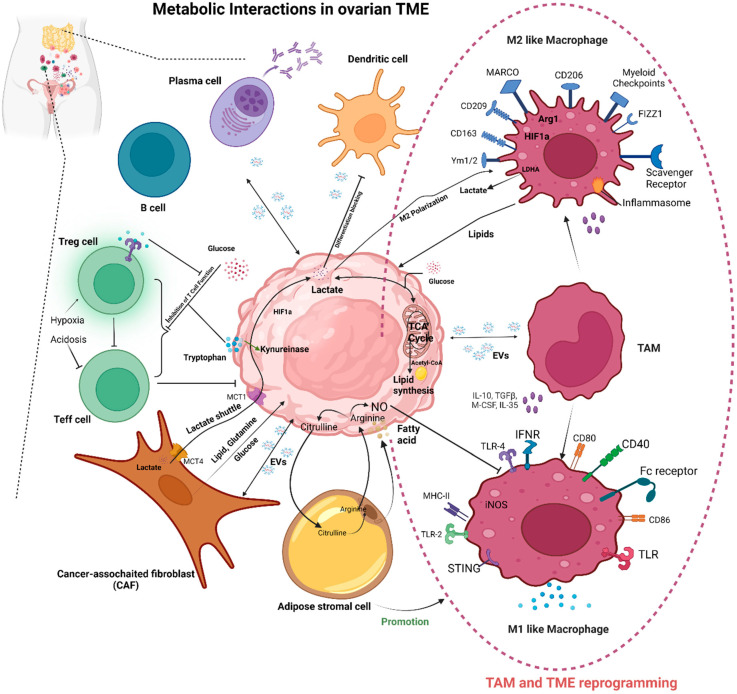
A summary of the metabolic interactions and reprogramming in the TME between cancer and non-cancer cells, focusing on TAM metabolic reprogramming, including EVs. Ovarian cancer cells consume oxygen and nutrients (glucose, glutamine, etc.) from the extracellular matrix, primarily through glycolysis, rather than through the TCA cycle of OXPHOS in the mitochondria, which provides ATP for cancer cell growth, even under aerobic conditions. During TAM and TME reprogramming, several signals, including complement, inflammasome activators, ligands for scavenger receptors MARCO and CD206, and myeloid checkpoints, can place macrophages in a pro-tumor mode. OXPHOS, oxidative phosphorylation; CAF, cancer-associated fibroblasts; Teff, effector T cells; Treg, regulatory T cells; Gln, glutamine; NO, nitric oxide; ARG1, arginase 1; MCT, monocarboxylate transporters; EVs, extracellular vesicles. Created with BioRender.com. Accessed on 11 September 2022.

## Data Availability

Not applicable.

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
