# Peer review of "Metabolic Reprogramming in Tumor-Associated Macrophages in the Ovarian Tumor Microenvironment"

_cancers, 2022, doi:10.3390/cancers14215224_

Round 1

Reviewer 1 Report

Kumar et al. intended to explain how macrophage immuno-metabolism changes macrophage phenotypes and activities in tumor microenvironments in their study titled "Metabolic Reprogramming In Tumor-Associated Macrophages In The Ovarian Tumor Microenvironment." The application of metabolic reprogramming in TAM for therapeutic intervention and drug resistance in ovarian cancer has also been covered in this review article.

This constitutes a large, comprehensive, and broadly rational body of review work that is appreciable. However, there are a number of minor concerns that can be resolved to improve the quality of the manuscript, listed below.

1. Regarding the metabolic change in the tumor microenvironment, please consider incorporating a brief description of the glycolysis route in addition to the OXPHOS pathway.

2. When discussing Metabolic Reprogramming, it is important to consider including the Warburg impact on the Ovarian Tumor Microenvironment.

Author Response

We have addressed all the comments raised by the reviewers. Please find the response to the reviewer's comments in the attached file. 

Thank you

Reviewer 2 Report

General Comment

Kumar et al present metabolic reprogramming of the TAM and TME in the progression and tumour growth of ovarian cancer. This is a unique perspective by focusing on the major metabolic pathways and determinants regulating the macrophage immune metabolism  and the phenotypic functions of the macrophages in the ovarian TME. The following are my comments and critique:

Simple summary

Since it is a brief summary, it is necessary to reduce the meaningless connecting words and directly summarize the important content.

Abstract

This is too wordy. We do not need to read a litany of information in the abstract. Just strive to summarize the most important content in concise language. And in line 2, the authors introduce the term tumor microenvironment for the first time, and the abbreviation need to be written here.

Introduction

The significance of macrophages for cancer cells should be clearly stated and extended. Because the article mainly focused on metabolic changes associated with TAM regarding disease progression and treatment of ovarian cancer. More literature and references are needed here to show the relation between metabolism and macrophages.

2.1. Glucose metabolism

Paragraph 4: “VEGF”, “PDGF” should be mentioned as abbreviation in parentheses.

2.2. Fatty acid and Lipid Metabolism

There are several metabolic pathways and metabolites that influence macrophage polarization. Add more references to the literature when you talk about the pro-tumorigenic polarization of TAMs associated with ovarian cancer.

2.3. Amino Acid Metabolism

Paragraph 1: Literature improvement is still needed. Please compare your point of view with some recent work, especially the metabolic crosstalk between TAMs and ovarian cancer.

Paragraph 2: The addiction of ovarian tumour to glutamine is considered a key metabolite for cancer cells. Due to its special features and properties metabolically speaking, please also try to compare some other metabolites in ovarian tumour.

2.4. Metabolic reprogramming by metabolite shuttling between TAM and cancer cells

Paragraph3-7: Please rearrange and check the logical order in paragraph writing. Please also reorganise the sentences that could make them easier for readers to follow.

3. Conclusions and Perspective

The reviewer raised questions on the possibility of targeting SDH deficient mutations tumors  by exploiting their metabolic state. However, the reviewer ended abruptly. Interpretation could be deepened. Authors may consider to write conclusion and perspective separately. Pick the your battles.

Author Response

Please find the response to the reviewer's comments in the attached file. 

Thank you

Round 2

Reviewer 2 Report

revision adequate an appropriate